# The Role of Polo-Like Kinase 1 in Regulating the Forkhead Box Family Transcription Factors

**DOI:** 10.3390/cells12091344

**Published:** 2023-05-08

**Authors:** Xavier T. R. Moore, Lilia Gheghiani, Zheng Fu

**Affiliations:** 1Department of Biology, Virginia Commonwealth University, Richmond, VA 23284, USA; 2Department of Biochemistry and Molecular Biology, Virginia Commonwealth University, Richmond, VA 23298, USA; 3Department of Human and Molecular Genetics, VCU Institute of Molecular Medicine, Massey Cancer Center, Virginia Commonwealth University, School of Medicine, Richmond, VA 23298, USA

**Keywords:** FOX family, transcription factor, oncogene, tumor suppressor, targeted therapy

## Abstract

Polo-like kinase 1 (PLK1) is a serine/threonine kinase with more than 600 phosphorylation substrates through which it regulates many biological processes, including mitosis, apoptosis, metabolism, RNA processing, vesicle transport, and G_2_ DNA-damage checkpoint recovery, among others. Among the many PLK1 targets are members of the FOX family of transcription factors (FOX TFs), including FOXM1, FOXO1, FOXO3, and FOXK1. FOXM1 and FOXK1 have critical oncogenic roles in cancer through their antagonism of apoptotic signals and their promotion of cell proliferation, metastasis, angiogenesis, and therapeutic resistance. In contrast, FOXO1 and FOXO3 have been identified to have broad functions in maintaining cellular homeostasis. In this review, we discuss PLK1-mediated regulation of FOX TFs, highlighting the effects of PLK1 on the activity and stability of these proteins. In addition, we review the prognostic and clinical significance of these proteins in human cancers and, more importantly, the different approaches that have been used to disrupt PLK1 and FOX TF-mediated signaling networks. Furthermore, we discuss the therapeutic potential of targeting PLK1-regulated FOX TFs in human cancers.

## 1. Introduction

Polo-like kinases (PLKs) belong to a family of serine/threonine kinases that are conserved across a broad range of eukaryotic organisms, including *C. elegans*, mice, and humans [1,2]. The prototypical member of this family, polo, was initially identified in *Drosophila* in 1988, with two mutants of the *polo* gene subsequently studied [3]. One mutant was found to be lethal in early embryogenesis; the other was partially lethal but caused aberrant mitotic characteristics, such as multipolar mitotic spindles, abnormal centrosome structure, and circular arrangement of chromosomes [3]. The latter mutant additionally caused defects in chromosome segregation and polyploid cells, highlighting an essential role of polo kinase in cell division and maintenance of diploidy of larvae during development [3].

In mammals, five paralogues of *polo* have been identified, including *Plk1*, *Plk2*, *Plk3*, *Plk4*, and *Plk5*, each having functions in the cell cycle [4]. PLK1 is essential in early embryonic development and has a fundamental role during mitosis, including regulation of mitotic entry and cleavage furrow invagination, among many others [5,6,7]. PLK2 and PLK4 have been identified to play key roles in centriole duplication and biogenesis, respectively, while PLK3 appears to have regulatory roles in the G_1_/S and G_2_/M-phase transitions and DNA replication. PLK5 exhibits the greatest evolutionary divergence in the family and appears to be related to proliferative arrest due to its accumulation in quiescent cells. Additionally, PLK5 has functions in neuronal differentiation [4,8,9]. Herein, we will focus on PLK1, which has generated significant interest as a therapeutic target due to its overexpression in many human cancers and its association with higher tumor grade and poorer patient outcomes [10,11].

As a kinase, PLK1 phosphorylates substrates, altering their stability, localization, and activity. Proteomic screening has identified more than 600 proteins that interact with PLK1 in a phosphorylation-dependent manner during mitosis [12]. Characterization of these interacting proteins using mass spectrometry and Gene Ontology biological categories identified additional interactions with proteins relevant to DNA replication, DNA damage response (DDR), G_2_ DNA damage checkpoint recovery, apoptosis, metabolism, RNA processing, and vesicle transport [12]. The approach employed by Lowery and colleagues characterized mitosis-specific interactions; however, additional studies have identified interactions outside of mitosis, including those with proteins involved in DNA replication, DDR, DNA damage checkpoint recovery, and telomere stability [13,14]. In human cancers, PLK1 has been identified to interact with many proteins involved in apoptosis, autophagy, metabolism, inflammation, epithelial-mesenchymal transition, and tumor invasion [13,15]. Interestingly, a global Plk1-overexpression murine model demonstrated that Plk1 overexpression promotes chromosomal instability, contributing to aneuploidy and spontaneous tumor formation [16]. Additionally, PLK1 also directly regulates numerous transcription factors (TFs), including repressor element 1 (RE-1)-silencing transcription factor (REST), p53, and members of the forkhead box (FOX) family of TFs (FOX TFs) [17,18,19,20]. Such discoveries have significantly increased our understanding of the PLK1 regulatory network and supported the notion that PLK1 is a master kinase coordinating diverse cellular processes. In this review, we focus on the role of PLK1 in regulating FOX TFs.

## 2. Structure and Regulation of PLK1

### 2.1. Structure of PLK1

PLK1 protein consists of an N-terminal kinase domain (KD) and a C-terminal polo-box domain (PBD) (Figure 1A). The KD is required for PLK1 phosphorylation activity, while the PBD is essential for proper protein localization, substrate interactions, and regulation [7].

The PLK1 KD contains the 11 subdomains typical of kinase catalytic domains, as well as the consensus sequences D-L-K-L-G-N and G-T-P-Y-I-A-P-E in the subdomains VIb and VIII, respectively, which indicates that PLK1 is a serine/threonine kinase [27,28]. The subdomains VII and VIII contain the residues forming the T-loop, which includes the essential threonine T210 [28]. Quantitative phospho-proteomics has defined a set of PLK1 consensus sequences, including the D/E-X-[S/T]-ψ-X-D/E motif, where X is any amino acid and ψ is a hydrophobic amino acid [29]. Additional variations of this motif have been described with the requirement for either an asparagine in the −2 position (L-N-X-[S/T] or N-X-S/T) or a phenylalanine in the +1 position ([S/T]-F) [29,30].

Of note, the PLK1 catalytic domain contains motifs unique to the PLK family, specifically several amino acid substitutions within the ATP-binding pocket [31]. For catalytic activity, ATP binds in a cleft formed by two lobes (residues 37–131 and residues 138–330) linked by a conserved hinge region in which key interactions, such as with K82, occur (Figure 1B) [31,32]. The ATP-competitive inhibitor of PLK1, BI 2536, leverages the amino acid substitutions within the ATP-binding pocket for specificity over many kinase families [22,33,34]; however, the homology of the ATP-binding pocket within the PLK family results in similar IC_50_ values for PLK1-3 [33,34].

The C-terminal region of PLK1 contains the PLK family-specific polo-box domain (PBD), which is essential for proper protein localization and substrate recognition [35]. Within the PBD are two polo boxes, polo-box 1 (PB1) and PB2, which, although sharing only 12% of sequence identity, form very similar three-dimensional structures, each containing a six-stranded antiparallel β-sheet and a singular α-helix (Figure 1C) [35,36]. An N-terminal extension covering residues 372 to 410, called the polo-cap, wraps around PB2 and stabilizes the overall three-dimensional structure of the PBD [36].

Investigation of the optimal PBD-binding motifs has revealed that the PBD preferentially binds to the motif S-[pS/pT]-P/X, with a strong selection for serine at the −1 position and a weaker selection for proline at the +1 position [37]. To generate the optimal phosphorylated motif, PLK1 substrates are frequently primed by the activity of cyclin-dependent kinases (CDKs). However, this is not exclusive [38]. Interestingly, PLK1 may also phosphorylate substrates through a self-priming mechanism or bind substrates in a phosphorylation-independent manner [39,40]. Characterization of the interface between the PBD and substrates has revealed that substrates bind in a shallow groove between PB1 and PB2, forming key interactions with four conserved residues, W414 and L490 in PB1 and H538 and K540 in PB2 [35,36,41]. The residues W414, H538, and K540 form key interactions with the threonine/serine phosphate group in the optimal phosphopeptide, while L490 interacts with the C-terminal portion of the phosphopeptide through non-polar interactions [35,36]. Interaction between the PBD and a substrate facilitates the interruption of intramolecular inhibitory interactions and provides a mechanism for the spatiotemporal regulation of PLK1 activity [36].

### 2.2. Regulation of PLK1

#### 2.2.1. Regulation of PLK1 Expression

As a result of its broad functions, PLK1 expression and activity are tightly regulated and vary dramatically during progression through the cell cycle [42,43]. In G_1_ and S-phases, PLK1 levels are low. However, as cells enter the G_2_/M phase, PLK1 levels dramatically increase before a rapid turnover at the onset of anaphase [43]. A tightly choregraphed balance of *PLK1* transcription, protein turnover, and cyclin-dependent kinase (CDK)-regulated activation mediates this dramatic fluctuation in PLK1 expression and activity.

The promoter region of *PLK1* contains multiple regulatory elements, including a putative SP1 element, a CCAAT box, and cell cycle-dependent element, and cell cycle genes homology region (CDE-CHR) sequences [44,45]. Cell cycle-dependent expression of *PLK1* is predominately mediated by the activity of E2F complexes [46]. However, there is also evidence indicating a role for FOX TFs in regulating *PLK1* expression. Previous reports have indicated that FOXM1 directly binds to the CHR consensus sequence, thus promoting cell cycle-dependent *PLK1* expression [47,48]. FKHRL1 (FOXO3 in the standardized nomenclature) binds to the promotor of *PLK1*, thereby upregulating its expression [49]. These data demonstrate the feedback loops between FOX TFs and PLK1, and further studies may identify other FOX TFs with regulatory control over *PLK1*.

#### 2.2.2. Regulation of PLK1 Stability

Initially, PLK1 adopts an inactive conformation in which the PBD binds to the KD, masking a nuclear localization sequence (NLS), enforcing PLK1 cytoplasmic localization, and reducing substrate interactions [50]. PLK1 is activated by the phosphorylation of T210 in the PLK1 T-loop by Aurora A [51,52]. Activation of PLK1 spikes approximately 40 min before the onset of mitosis and commits the cell to mitotic entry [53].

PLK1 turnover is mediated by ubiquitination and requires the presence of a D-box-like motif, R-K-P-L-T-V-L-N-K, in which the residues R337 and L340 are essential for proteasomal-mediated degradation of PLK1 [54]. During interphase, PLK1 turnover is mediated by SKP1/CUL1/F-box protein (SCF) E3 ubiquitin ligase complexes, while the activity of anaphase-promoting complex/cyclosome (APC/C) is required for mitotic exit [54,55,56].

## 3. Regulation of the FOX Family by PLK1

In humans, FOX TFs encompass a diverse family of approximately 50 TFs divided into 19 subfamilies (FOXA through FOXS) [57,58]. These proteins regulate a broad range of molecular cascades through which they exert influence upon cellular proliferation, differentiation, metabolism, senescence, and apoptosis from development through adulthood [59,60]. The three-dimensional structure of the FOX TF DNA-binding domain (FOX-DBD) is used to organize these TFs into subfamilies and is highly conserved across all FOX TFs [61]. The FOX-DBD consists of approximately 100 amino acids that form 2 loop structures from 3 α-helices and 3 β-sheets, creating a structure reminiscent of butterfly wings [62]. The differences in the regulation and structural domains present in FOX TFs afford the high specificity to their respective high-affinity binding sites in vivo. However, a general consensus sequence of 5′-(G/A)-(T/C)-(A/C)-A-A-(C/T)-A-3′ has been identified [63].

The transcriptional activity of FOX TFs regulates a broad range of developmental processes, and alterations in a single FOX TF frequently cause severe phenotypes or embryonic lethality [60]. As a result of their importance in development, dysregulation of FOX TFs is frequently observed in human cancers. Members of the FOXM, FOXO, FOXK, and FOXC subfamilies are key members of both oncogenic and tumor suppressive pathways, with significant regulatory control over tumorigenesis, tumor progression, and therapeutic resistance [59]. As a result, there is significant interest in understanding the post-translational modifications (PTMs) that regulate the stability, localization, and activity of FOX TFs, as well as the networks of interactions through which these TFs are modified [57,64,65]. To this end, proteomic analysis of FOX TFs has found distinct differences in the composition of protein complexes and PTMs between soluble and chromatin-bound fractions [66].

### 3.1. FOXM1

#### 3.1.1. Background

FOXM1 was initially identified in mitotic HeLa cells and named MPM2-reactive phosphopeptide 2 (MPP2) due to its reactivity with the mitotic phosphoprotein antibody, MPM2 [67,68]. Follow-up studies identified the presence of this protein in proliferating colon cancer [69], mouse thymus [70], and rat insulinoma cells [71]. Prior to the standardization of FOX TF nomenclature [61], FOXM1 was referred to by numerous names, including Trident (in mice); Win or ISN-1 (in rats); and MPP2, FKHL16, HNF-3, or HFH-11A/B (in humans) [72]. Naming was further complicated by the presence of isoforms arising from alternative splicing [65,73].

FOXM1 consists of an N-terminal repressor domain (NRD), a conserved forkhead DBD, and a C-terminal transactivation domain (TAD) encoded by a gene containing 10 exons. Alternative splicing of exons Va and VIIa, alternatively referred to as A1 and A2, respectively, gives rise to three isoforms, FOXM1a, FOXM1b, and FOXM1c [65]. When expressed, exon Va is found at the C-terminus of the DBD, while exon VIIa is found within a linker between the DBD and TAD (Figure 2) [69,70,73]. Characterization of the different FOXM1 isoforms has revealed that in both normal and tumor tissues, FOXM1a has the lowest expression while FOXM1c has the highest expression. Interestingly, all isoforms exhibit increased expression (of a similar magnitude) in tumor tissues compared to normal tissues, suggesting that the increased transcription observed in cancers is not accompanied by changes in post-transcriptional splicing [74]. Transcriptional activity reporter assays have revealed that FOXM1b and FOXM1c have comparable levels of activity, which are significantly higher than that of FOXM1a [74]. In a 293T CRISPR knockout model of *FOXM1*, differential gene expression analysis of *FOXM1-*null cells reconstituted with individual isoforms revealed both overlapping and unique target genes of FOXM1 isoforms [74].

#### 3.1.2. Functions

Initial efforts identified FOXM1 as having key roles during development and cell cycle regulation. Investigation of FOXM1 in human cancers has expanded its known functions to include the promotion of proliferation, EMT, invasion, angiogenesis, and cancer stem cell (CSC) phenotypes.

The essential nature of *FOXM1* was initially identified using knockout mice, revealing that *Foxm1*-null mice exhibited greater than 90% embryonic lethality. Surviving mice had severe hepatic and cardiac hypoplasia-induced defects as well as significant polyploidy [75,76,77]. Characterization of *Foxm1*-depleted mouse embryonic fibroblasts revealed G_2_ delays, chromosomal mis-segregations, compromised spindle assembly checkpoints, and cytokinesis defects [47]. Early investigation of FOXM1 target genes with DNA microarrays revealed that Foxm1 activation induces differential expression of numerous cell cycle genes, including those encoding multiple cyclins, CENP-F, Aurora B, and Plk1 [47]. FOXM1 depletion resulted in the accumulation of the CDK inhibitor proteins, p21^Cip1,^ and p27^Kip1^, within the nucleus due to reduced expression of SCF ubiquitin ligase complex components [78].

In cancer, FOXM1 is a key regulator of several cancer-promoting phenotypes. Ubiquitous *Foxm1* overexpression in mice resulted in the formation of a significantly greater number of tumors that were larger, had increased proliferation, and had greater DNA complements than wild-type (WT) mice [79]. In vitro, FOXM1 drives cancer cell proliferation in multiple tumor types, including liver, neuroblastoma, and prostate cancer (PCa) [80]. Additionally, FOXM1 overexpression promotes EMT, invasion/metastasis, and tumor-supportive angiogenic outgrowth. In a PCa model, TGF-β1-induced EMT was found to be reversible through the knockdown of FOXM1, which caused reduced expression of vimentin, SLUG, and Zeb2 [81]. Similar results had been obtained in hepatocellular carcinoma (HCC) and triple-negative breast cancer (TNBC) models in which FOXM1b dysregulation promoted the expression of mesenchymal markers, such as N-cadherin, Snail, and Zeb1, as well as the repression of the epithelial marker E-cadherin [82,83]. In glioma cells incapable of forming tumor xenografts, FOXM1b overexpression enabled tumor formation in nude mice and promoted invasion through *MMP-2* upregulation [84,85]. In TNBC and PCa models, FOXM1 was identified as the effector of receptor tyrosine kinase-induced EMT and in vitro invasiveness [81,83]. FOXM1 overexpression in HCC and colorectal cancer (CRC) models leads to cytoskeletal remodeling and increased metastasis in vivo [82,86]. In support of tumor growth, FOXM1b overexpression has been observed to result in significant angiogenesis through direct upregulation of *VEGF* [87]. Similar results were obtained in a TRAMP PCa mouse model in which deletion of *Foxm1* resulted in decreased expression of *Vegf-A* [88]. In breast cancer cell lines, active FOXO3 has been reported to displace FOXM1 from the *VEGF* promoter, thereby reducing *VEGF* gene expression [89].

Accumulating evidence indicates that reactivation of embryonic and pluripotency pathways in a subpopulation of tumor cells, CSCs, results in a stemness phenotype that enhances therapeutic resistance and cancer recurrence [90]. A growing pool of evidence indicates that FOXM1 plays a key role in promoting and maintaining CSC populations in multiple cancers. For example, in a non-small-cell lung cancer (NSCLC) model, isolated CSCs exhibited elevated FOXM1, the knockdown of which resulted in reduced expression of stem cell markers (CD133 and CD44), stem cell regulators (Bmi1, Sox2, and Oct4), and self-renewal [91]. In addition to lung cancer, FOXM1 has been identified as supporting a CSC phenotype in many other cancer types, including breast, colorectal, hepatic, and pancreatic cancers [92].

Additionally, FOXM1 has been identified as promoting therapeutic resistance in multiple treatment modalities across various cancers, including pancreatic, glioblastoma, and breast cancers [92]. In addition to the promotion of CSC phenotypes, the contribution of FOXM1 to therapeutic resistance appears to be partly due to the broad regulation of DNA damage response genes to increase DNA repair capacity, thereby rendering resistance to DNA damage-induced cytotoxicity [93]. The relationship between CSC phenotypes and therapeutic resistance combined with the identification of FOXM1-regulated DDR genes raises the interesting question that the therapeutic resistance observed in CSC populations may, in part, be due to high FOXM1 expression causing increased DDR capacity. Further investigation of these observations may reveal rational therapeutic combinations to increase the efficacy of targeting CSC and therapeutic-resistant populations.

#### 3.1.3. Regulation

FOXM1 is tightly regulated at the transcriptional and post-transcriptional levels, as well as post-translationally through both protein-to-protein interactions and PTMs regulating localization, activity, and turnover of FOXM1 [65]. In this review, we focus on the role of PLK1 in regulating FOXM1.

During G_1_, FOXM1 protein levels are low, and FOXM1 is generally inactive due to repressive inter- and intra-protein interactions [94,95]. Initial efforts exploring FOXM1 activation identified multiple CDK phosphorylation sites, the disruption of which resulted in the downregulation of FOXM1 target genes [96,97,98,99]. However, more recent studies have identified that the key event in FOXM1 activation is PLK1 phosphorylation at S715, which leads to the disruption of repressive NTD-TAD interactions and subsequent protein activation [19,95].

As previously noted, priming phosphorylation events by CDKs provide binding sites that are recognized by the PLK1 PBD to facilitate PLK1–substrate interactions. T596 and S678 have been identified as key FOXM1b residues that must be phosphorylated by CDK1 for PLK1 binding [19]. Occurring in late G_2_/M, the FOXM1-PLK1 interaction results in subsequent PLK1-mediated phosphorylation at S715 and S724, thus increasing FOXM1 target gene transactivation [19]. PLK1-dependent regulation of FOXM1b activity provides a positive feedback loop that drives increased PLK1 expression and FOXM1b activation, ensuring the execution of orderly mitotic progression [19]. As PLK1 expression and activity are regulated in a cell-cycle dependent manner, these regulatory phosphorylations also cyclically occur as the cell progresses through the cell cycle [19,42]. A follow-up study revealed that FOXM1b was SUMOylated in vitro and in vivo, resulting in increased cytoplasmic localization, increased proteolytic degradation, and reduced transcriptional activity [100]. PLK1-mediated phosphorylation of FOXM1b antagonizes SUMOylation via SUMO-1, thereby promoting FOXM1b nuclear translocation and transcriptional activity [100]. Taken together, these studies reveal the key role played by PLK1 in regulating the activity and stability of FOXM1b, which is essential for timely progression through mitosis.

### 3.2. FOXO1 and FOXO3

#### 3.2.1. FOXO1 and FOXO3 Background

The FOXO subfamily that includes FOXO1, FOXO3a, FOXO4, and FOXO6 is the most evolutionarily divergent clade of FOX TFs [101]. These TFs are differentially expressed in various tissues with diverse regulatory functions [102,103]. In vivo, these proteins display redundant tumor-suppressive functions, but, interestingly, have distinct lineage and organ-specific effects arising from the differential expressions of both unique and overlapping target genes [104].

*FOXO1* and *FOXO3* encode the proteins of 655 and 673 amino acids, respectively, which contain four major functional regions: a FOX-DBD, a NLS, a nuclear export signal (NES), and a C-terminal TAD (Figure 2) [105]. In contrast to FOXM1, which activity is regulated primarily by NRD-TAD interactions, FOXO activity is predominantly regulated by nuclear–cytoplasmic shuttling and proteolysis [106].

#### 3.2.2. FOXO1 and FOXO3 Functions

Characterization of FOXO proteins has revealed their important roles in the regulation of apoptosis, cell cycle progression, and cellular homeostasis. These broad functions situate these proteins within the regulatory networks associated with numerous hallmarks of human cancers [107].

The FOXO family plays key roles in apoptosis and cell cycle arrest [108,109]. Interrogation of the molecular functions of FOXO1 has identified it as a key regulator of pro-apoptotic genes, including *Bim*, *Puma*, and *FasL* [110]. Overexpression of either FOXO1 or FOXO3 rapidly induced apoptosis in LAPC4 cells through the upregulation of several pro-apoptotic genes, including *TNFSF10* (*TRAIL*), *DAPK1*, and *BNIP3L* [111]. In addition, FOXO1 and FOXO3 also control the expression of several regulators of cell cycle progression. For instance, FOXO1 downregulates the expression of *cyclins D1* and *D*2 while upregulating the expression of *cyclin G2* and tripling the half-life of *p27^KIP1^*, thereby causing G_1_ arrest [112,113]. In response to DNA damage, FOXO1 promotes the transcription of *p27^KIP1^* and *GADD45*, resulting in cell cycle arrest [114]. Additionally, FOXO3 downregulates the expression of *cyclin B1* and *CDCA3* while upregulating the expression of *cyclin G2*, which contributes to cell cycle arrest. FOXO3 has been identified to downregulate key S-phase genes *cyclin A2*, *CDC45L*, and *MCM3* [115].

FOXO proteins have been identified as playing key roles in maintaining cellular homeostasis through their regulation of metabolism and oxidative stress. FOXOs regulate several genes essential for glucose and lipid metabolism; however, FOXO1 and FOXO3 appear to have distinct but related functions in which FOXO1 has a greater role in mediating insulin responses, while FOXO3 regulates metabolic flux to maintain redox homeostasis [116,117]. Additionally, FOXOs have been identified as upregulating a multitude of antioxidant genes, including those encoding members of the catalase, superoxide dismutase, and peroxiredoxin families in response to activation by stress-response kinases [116,118]. Whether FOXO proteins promote the expression of antioxidant or pro-apoptotic genes in response to oxidative stress appears to be, at least in part, dependent upon the degree of acetylation of FOXO proteins [116].

In addition, FOXO proteins have been identified as playing important roles in preserving the cellular homeostasis required for the maintenance of stem cell populations. The loss of FOXO proteins in NSC and hematopoietic stem cell populations has been identified as resulting in reduced self-renewal and population collapse due to increased oxidative stress [119,120]. In human embryonic stem cells, FOXO1 overexpression upregulates several pluripotency markers, including *OCT4*, *SOX2*, *NANOG*, and *KLF4* [117]. Findings that FOXO1 and FOXO3 support stemness phenotypes in cancer cells likely have clinical significance and warrant further investigation [121,122].

FOXO proteins are generally considered to be tumor suppressors due to their negative regulation of cell cycle progression, positive regulation of apoptosis, and their role in cancer-related metabolic dysregulation [101,123]. *FOXO1* knockdown has been reported to result in enhanced proliferation, motility, and invasive potential, as well as increased EMT-related gene expression in HCC, NSCLC, and PCa cell lines [109,124,125]. In vivo, FOXO1 overexpression reduced lung metastasis in number and size compared to controls [124]. In a mouse PCa model, *Erg* overexpression was insufficient to cause pathology; however, when combined with *Foxo1* knockout, more than 50% of mice exhibited high-grade prostatic intraepithelial neoplasms [109]. Tumor-suppressive functions of FOXO3 have been reported. For instance, FOXO3 activation promotes apoptosis in many cell lines, including those for breast cancer, oral squamous cell carcinoma, osteosarcoma, gastric cancer, and ovarian cancer [126,127,128,129,130]. Cell motility assays in urothelial cancer models revealed that *FOXO3* knockdown resulted in increased mobility, an effect that is reversible with simultaneous *TWIST1* knockdown [131]. Additionally, dysregulation of FOXO3 has been identified to result in accelerated tumor formation and disease progression in TRAMP mice [132]. However, more recent evaluation suggests that the simple classification of FOXO proteins as tumor suppressive fails to encompass the functions of these proteins; a combination of in vitro and clinical data indicates that these proteins can in fact have pro-tumor functions [133]. Accumulating evidence suggests that the broad homeostatic-promoting functions of FOXO proteins are present in both normal and neoplastic cells and, dependent upon the molecular milieu, can promote or repress tumorigenesis. The complex role of FOXO proteins in cancer initiation and progression has been discussed in detail elsewhere [133,134].

#### 3.2.3. FOXO1 and FOXO3 Regulation

In addition to transcriptional and post-transcriptional regulation, FOXO proteins are subject to numerous PTMs regulating their expression, localization, and activity [64]. Their localization and transcriptional activity appear to be most heavily influenced by phosphorylation, while acetylation plays an important role in fine-tuning FOXO transactivation [105,106,116]. Isoform-specific PTMs and the unique protein–protein interactions resulting from these PTMs contribute to the distinctive phenotypes observed in null mice [64,135]. FOXO proteins are regulated by numerous kinases, and of particular interest herein is the negative regulation of FOXO proteins by PLK1 [20,136].

An RNAi screen of kinases and phosphatases with regulatory effects upon dFoxO in *Drosophila* identified polo [137]. An investigation in mammalian cells revealed that PLK1 phosphorylates the evolutionarily conserved residues, S75, S218, and S329, in late G_2_/M. Interestingly, PLK1 acts as the priming kinase by phosphorylating S218 and S329 to create docking sites to facilitate subsequent phosphorylation at S75 [20]. Characterization of its functional significance revealed that PLK1-mediated phosphorylation promoted nuclear exclusion and reduced FOXO1 transcriptional activity, impairing both FOXO1-induced apoptosis and G_2_/M delay, which enables the normal execution of the cell cycle program [20]. These data indicate the cell cycle-dependent regulation of FOXO1 by PLK1 and provide insight into how PLK1 antagonizes cell cycle arrest and apoptosis.

Tandem affinity purification and mass spectrometry performed by Bucur et al. identified FOXO3 as a novel PLK1 interaction partner. The PLK1/FOXO3 interaction is mediated by a fragment containing residues 219–270, which includes part of the FOX-DBD and the NLS [138]. Further studies have revealed that PLK1 phosphorylates FOXO3 in vitro and that overexpression of PLK1 promotes nuclear exclusion, enhanced degradation, and reduced FOXO3 target gene expression [138]. Further investigation to determine the mechanism of in vivo regulation of FOXO3 and the functional outcomes of PLK1 interactions with FOXO3 may reveal novel PLK1-regulated networks arising from these unique FOXO3 functions.

### 3.3. FOXK1

#### 3.3.1. Background

The FOXK subfamily has only recently begun to gain significant research interest, following the recognition of its important roles in human cancers [139,140]. The FOXK subfamily was initially identified in mice as myocyte nuclear factor (MNF), where initial characterization revealed transient expression in some early embryonic tissues, but persistent expression in quiescent satellite cells [141,142]. Later, in silico studies of the human genome identified a gene encoding a FOX protein with significant homology to the murine MNF, which was subsequently named FOXK1 [143]. More recent tissue-based mapping of the humane proteome has revealed the broad expression of FOXK1 in many human tissues [144].

The FOXK1 protein contains a FOX-DBD and a forkhead-association (FHA) domain, which mediates phospho-dependent protein-to-protein interactions [145]. In addition, the N-terminal contains a Sin3-interacting domain (SID), which mediates interactions with SWI-independent-3 (Sin3), a histone deacetylase-containing repressor complex, to regulate cell cycle progression in myogenic progenitor cells (Figure 2) [146]. Differential splicing of *FOXK1* results in two isoforms with differing lengths, FOXK1a and FOXK1b [143]. To date, the difference between these isoforms in humans has not been reported; however, similar to FOXM1, they likely have both unique and overlapping targets. Interestingly, in a mouse model of muscle regeneration, the two mouse isoforms, MNF-α and MNF-β, exhibited differential expression in actively proliferating myoblasts and quiescent satellite cells [147].

#### 3.3.2. Functions

Characterization of FOXK1 has identified its functions in multiple cancer hallmarks, including proliferation, metastasis, angiogenesis, and apoptosis, as well as its roles in metabolism [107].

An investigation of the functions of FOXK1 in human cancers revealed that it promoted proliferation, EMT, invasion, and metastasis across multiple human cancers, including glioma, ovarian cancer, gastric cancer, and CRC [145]. In glioma, FOXK1 overexpression promoted proliferation through the upregulation of cyclin D, c-myc, and β-catenin; promoted progression into S-phase; and reduced apoptosis [148]. Similar results have been obtained in ovarian cancer models in which FOXK1 overexpression induced invasion and migration, indirectly increased the expression of *MMP-9*, and directly repressed the expression of *p21*. In gastric cancer cells, a TGF-β-induced increase in proliferation, migration, EMT, and metastatic potential was found to be dependent upon the upregulation of *FOXK1* by c-jun. In vivo, FOXK1 overexpression increased tumor volume, microvessel density, and metastasis [149]. Interestingly, FOXK1 overexpression in a cellular model of CRC resulted in increased transcription of multiple oncogenes, including cyclin D1, β-catenin, and Myc as previously identified, but also ZEB1, ID1, Sp1, TWIST, TERT, and survivin. Further studies revealed that siRNA knockdown of FOXK1 resulted in the activation of several caspases and potentiated cells to apoptotic stimuli in both in vitro and in vivo systems [150].

Additionally, FOXK1 has been identified as a key effector of AKT-mTOR signaling cascades and, in response to metabolic signals, regulates the expression of genes related to biosynthetic, metabolic, autophagic, and stress-response pathways [151,152,153]. These functions are beyond the scope of this review but have been discussed at length by others [145].

#### 3.3.3. Regulation

FOXK1 localization, transcriptional activity, and stability are predominantly regulated through phosphorylation [145]. Phosphoproteomics has identified more than a dozen differentially phosphorylated serine and threonine residues that alter FOXK1 localization and activity [151,154]. FOXK1 nuclear–cytoplasmic shuttling is regulated by the mTOR-Akt signaling axis via similar mechanisms as FOXO proteins, albeit with opposing outcomes [152]. Specifically, it is regulated by GSK3 and PP2A in which GSK3 phosphorylation promotes its cytoplasmic sequestration, and mTOR-dependent PP2A phosphatase activity promotes its nuclear accumulation [151,154,155].

There is emerging evidence suggesting that PLK1 plays a role in regulating FOXK1 activity. Ramkumar and colleagues reported that PLK1 phosphorylates the scaffold protein JLP at T351, creating a PLK1-binding site and enabling further C-terminal phosphorylation of JLP by PLK1, which then facilitates the recruitment of FOXK1 and FOXK2 to JLP during G_2_/M. Further studies revealed that FOXK1 interacts with JLP via the FOXK1 FHA domain binding to the previously identified residue, T351, on JLP. PLK1 also directly interacts with the FOXK1 DBD to form a JLP-FOXK1-PLK1 complex. Interestingly, the knockdown of JLP with shRNA resulted in increased FOXK1 stability [156]. Ramkumar and colleagues stated, albeit without evidence, that FOXK1 is an in vitro substrate of PLK1. The broad cancer-promoting functions of FOXK1 warrant further investigation to characterize the significance of this colocalization of PLK1 and FOXK1 and to determine what role it may play in tumor progression. It is possible that PLK1 may phosphorylate FOXK1 in vivo to promote ubiquitination and proteolysis; however, further investigation is required to validate this hypothesis.

### 3.4. Other FOX Transcription Factors

Evidence suggests that FOXC2 may be a PLK1 substrate. In breast CSCs, FOXC2 regulates the expression of key cell cycle genes and, notably, has an evolutionarily conserved putative PLK1 site at S125 [157]. Further investigation revealed that inhibition of PLK1 by BI 2536 resulted in increased FOXC2 turnover, which was reversible by cotreatment with MG132, suggesting a stabilizing function of PLK1. Interestingly, enhanced FOXC2 expression increased sensitivity to PLK1 inhibition in TNBC CSCs; however, the precise mechanism remains to be defined [157].

## 4. The Clinical Significance of PLK1 and FOX Transcription Factors

### 4.1. PLK1 and FOX Transcription Factors in Cancer

The clinical significance of PLK1 has been recognized since the early 2000s. Studies of patient tumor samples have revealed a positive association between *PLK1* expression and histopathologic grading in numerous human cancers, including ovarian, prostate, and gastric cancers [158,159,160]. High PLK1 expression has additionally been associated with poorer patient outcomes in several human cancers, including lung cancer, head and neck squamous cell carcinoma, and oropharyngeal cancer [161,162,163]. The most thorough analysis of the prognostic significance of PLK1 has been conducted by Liu and colleagues, who analyzed RNA-seq data from the TCGA data set. Compared to normal tissues, *PLK1* was significantly overexpressed in 18 of 19 analyzed human cancers, with the magnitude of *PLK1* overexpression ranging from 2-fold to more than 20-fold [10]. Further analysis sought to determine the prognostic value of *PLK1* expression in 25 different cancer types, revealing a negative association between PLK1 expression and overall survival in 10 cancer types, including breast invasive carcinoma, lung adenocarcinoma, and pancreatic ductal adenocarcinoma [10]. In addition, an evaluation of disease-free survival in 26 cancer types demonstrated a negative association with *PLK1* expression in 7 cancer types [10].

While significant evidence indicates a connection between PLK1 and human cancer, recent reports have provided conclusive evidence of the in vivo oncogenic effects of PLK1. A genetically engineered mouse model evaluating the effects of global *Plk1* overexpression demonstrated that Plk1 dysregulation induced chromosomal instability and compromised cell cycle checkpoints by impairing p53 activity, leading to spontaneous tumor formation [16]. In a *Kras*^G12D^/*Tp53*^fl/fl^-driven lung adenocarcinoma (LUAD) mouse model, the addition of *Plk1* overexpression enhanced tumor burden and accelerated tumor formation through Ret upregulation, resulting in greater MAPK and PI3K signaling [164]. These mouse models provide compelling in vivo evidence of the oncogenic effects arising from PLK1 dysregulation and, in conjugation with other studies, indicate a potentially important role for PLK1 in repressing p53 activity [16,18]. Several studies have indicated a cooperative, pro-apoptotic FOXO3a-p53 network, while others have demonstrated that FOXM1 expression is repressed by p53 [165,166,167,168], suggesting that WT p53 may have an important role within the PLK1-FOX TF interplay [16,18,169]. As such, in the instance of cancer, where PLK1 is often both overexpressed and overactive, inactivating PLK1-p53 interactions may impair cooperative p53-FOXO3a interactions and/or the concomitant repression of *FOXM1*. Due to the importance of these proteins, further investigation is warranted.

In addition to PLK1, FOXM1 has also been identified as having significant prognostic value. A meta-analysis conducted by Li and colleagues analyzed 23 publications, including nearly 3000 patients across 14 different cancer types, and found that FOXM1 overexpression was associated with reduced overall survival at 3, 5, and 10 years and a more advanced TMN stage [170]. An effort to identify prognostic genes in human cancers analyzed the transcriptomic and survival data of approximately 18,000 patients across 39 different malignancies and found that *FOXM1* expression was the most frequently associated gene with adverse prognostic significance across the entire data set [171].

In NSCLC and bladder cancers, lower FOXO1 expression was observed in tumor tissues compared to normal tissues [125,172]. Further analysis of these bladder cancer samples revealed that lower FOXO1 expression was associated with higher clinical staging, greater lymph node metastasis, and poorer prognosis [172]. Investigation of FOXO1 inactivation (indicated by immunohistochemical staining for FOXO1 p-S256) in upper urinary tract urothelial carcinoma revealed that increased FOXO1 phosphorylation was observed in tumor tissue and was associated with increased invasiveness [173]. Paradoxically, FOXO1 was upregulated in esophageal squamous cell carcinoma (ESCC) tissue compared to normal tissue and was associated with reduced survival. Clinicopathological analysis identified a positive association between metastasis and FOXO1 expression, and multivariate cox regression analysis identified negative prognostic significance for FOXO1 in ESCC [174].

Of note, there is also evidence of prognostic significance for FOXO1 fusion proteins, mainly in tumors harboring PAX3-FOXO1 fusions. The PAX3-FOXO1 fusion protein arises from a commonly observed genetic abnormality of alveolar rhabdomyosarcoma (aRMS) that results from the fusion of the DBD of PAX3 and the TAD of FOXO1. A study of patient outcomes revealed that the PAX3-FOXO1 fusion protein (present in 55% of study patients) was associated with greater invasiveness and worse patient outcomes than fusion-negative patients or those harboring the PAX7-FOXO1 fusion protein (present in 22% of study patients) [175].

In urothelial cancers, FOXO3 expression is downregulated in invasive cases and is negatively associated with both disease-free survival and overall survival [131]. In another study, Shou and colleagues reported that low expression of FOXO3 was associated with poor clinical stage, increased metastasis, and poor clinical outcomes in nasopharyngeal carcinomas (NPC) [176]. However, a meta-analysis of the prognostic significance of FOXO3 in approximately 1000 HCC patients revealed positive associations between FOXO3 expression and tumor formation and invasiveness, as well as a negative association between expression and survival [177].

Studies in ovarian cancer have shown that FOXK1 is overexpressed and positively associated with poorer patient prognosis [178]. Similarly, an analysis of gastric cancer samples revealed increased expression of FOXK1 in tumor tissue compared to normal tissue [149]. In addition, FOXK1 is overexpressed in HCC, positively regulates a pluripotency network, and is associated with disease progression and poorer outcomes [179]. Further investigation into other human cancers may reveal whether FOXK1 has prognostic value in a broader range of human cancers than is currently understood.

### 4.2. PLK1 Inhibition

Owing to the critical role of PLK1 in human cancers, there has been significant interest in developing PLK1 inhibitors since the early 2000s. Initial efforts to develop inhibitors targeted the PLK1 ATP-binding pocket within the kinase domain; however, following the recognition of the limitations in this approach, more recent efforts have shifted focus to targeting the PBD.

To date, several ATP-competitive inhibitors of PLK1 have been explored for clinical application. These agents and their respective stages of clinical development have been explored in detail by Novais and colleagues [11]. However, PLK1 inhibitors have generally had limited clinical utility due to their modest effects as monotherapy agents and off-target toxicity caused by homology within the PLK family [11,180]. Additionally, in vitro evidence suggests that persistent exposure to BI 6727 (a development of BI 2536 with improved pharmacokinetic properties) results in the development of resistant phenotypes through increased expression of ABC transporters [181]. These data indicate the need for further development of more specific PLK1-targeted agents and the exploration of combination regiments to improve clinical responses and reduce the risk of resistant phenotypes.

Alternative approaches for PLK1 inhibition have sought to disrupt PLK1–substrate interactions with inhibitors binding to the PLK1 PBD. The first molecule leveraging this approach, called Poloxin, was reported by Reindl and colleagues in 2008, and was found to induce mitotic arrest and mitosis in HeLa cells [182]. However, the authors noted a relationship between time and efficacy, suggesting a role for covalent interactions in its mechanism of action. This speculation was confirmed by Archambault and colleagues, who identified Poloxin as a nonspecific protein alkylator, rendering the compound unsuitable for clinical development [183]. Since then, two improved compounds, Poloxin-2 and Poloxin-2HT, have been reported [184,185]. However, both compounds retain the activated ester motif suspected to be responsible for the alkylating activity of the original molecule. As a result, it is unlikely that any of these compounds are suitable for clinical development. While PBD-targeting inhibitors represent an improvement over competitive ATP-binding domain inhibitors, homology in the PBD of PLK proteins represents a persistent challenge for PLK1-specific drugs.

Recently, there have been attempts to specifically disrupt protein–protein interactions with agents targeting the shallow binding pockets on the surface of proteins. Novel approaches to develop high-affinity agents targeting PLK1 have explored agents targeting both the binding pocket and cryptic pockets (only revealed after ligand binding). Hymel and colleagues had leveraged this approach to develop several novel macrocyclic peptide mimetics with high affinity for the PLK1 PBD [186]. Characterization of these agents revealed nanomolar IC_50_ values and 140- and 300-fold specificity over the PBDs of PLK2 and PLK3, respectively [186]. Both the IC_50_ and specificity represent significant improvements over the micromolar and <100-fold selectivity of other PBD targeting agents [182,184,185].

Another therapeutic approach being explored is proteolysis-targeting chimeras (PROTACs), which bring a protein of interest (POI) and an E3 ubiquitin ligase in proximity. This proximity is created by combining moieties with an affinity for the POI and the E3 ligase via a linker, which results in POI degradation following ubiquitination [187,188]. Mu and colleagues developed a dual bromodomain 4 (BRD4) and PLK1 PROTAC system using a BI 2536 moiety for targeting. Evaluation of this PROTAC, HBL-4, in a model of acute myeloid leukemia (AML) demonstrated rapid degradation of both BRD4 and PLK1. Additionally, the HBL-4 induced greater apoptosis and similar tumor regression in tumor xenograft models at lower concentrations than BI 2536 [189]. The previously discussed limitations of BI 2536 raise the concern that this PROTAC may additionally cause degradation of PLK2 and PLK3, producing the same off-target effects of BI 2536. While this has yet to be investigated, these concerns could be addressed by using moieties with greater PLK1 specificity.

The toxicities arising from off-target effects of inhibitors and modest effects as monotherapy agents have limited the clinical benefit of targeting PLK1 [11]. However, significant pre-clinical data have demonstrated the potential of PLK1 inhibition and suggest that further development of high-specificity agents, particularly those targeting the PBD, may offer greater efficacy [190]. With the long-understood limitations of monotherapies, the characterization of oncogenic pathways and the role of PLK1 and PLK1 substrates may inform rational combination therapies.

### 4.3. Disruption of PLK1-Regulated FOX Transcription Factor Signaling

Due to the significant role played by FOXM1 in many human cancers, disruption of FOXM1 signaling could be an effective anti-cancer treatment strategy [191]. The difficulty with targeting TFs has resulted in few efforts seeking to disrupt FOXM1 functions. To date, the only therapies directed at FOXM1 have been a few cancer vaccines, which include FOXM1 peptides. Evaluation of these vaccines in early clinical trials with cervical, ovarian, gastric, and refractory pediatric solid tumors has found them to be tolerated with manageable side effect profiles and appear to positively affect patient outcomes [192,193,194,195]. With advances in drug development, targets previously thought to be undruggable, such as TFs, are now potential targets. The phosphorylation of FOXM1b at T596 and S678 by CDKs and subsequently at S715 and S724 by PLK1 generates unique motifs that could be explored as potential targets for drug development. Additionally, the necessity for PLK1-mediated phosphorylation for FOXM1 activation suggests that combining an anti-PLK1 therapy with an anti-FOXM1 therapy could potentially limit the FOXM1-driven pro-growth phenotypes and disrupt the PLK1-FOXM1 positive feedback loop that drives mitotic progression [19]. Combining these approaches with cancer vaccines may enable an immunochemotherapeutic approach with superior effects against cancer.

Increased expression of FOXO1-regulated pro-apoptotic proteins may have synergistic potential with other drugs. Building upon previous reports that PLK1 phosphorylation of FOXO1 promotes nuclear exclusion and degradation, Gheghiani et al. sought to determine if disrupting the PLK1-mediated inactivation of FOXO1 had therapeutic potential. The PLK1 inhibitor, BI 2536, was used in combination with the microtubule poison nocodazole across a range of concentrations. Low-dose combinations of BI 2536 and nocodazole proved to be cytotoxic to advanced PCa cells, while not significantly impacting normal prostate epithelial cells. The minimal effect on normal prostate epithelial cells suggests that such a combination has the potential to minimize side effects [196]. These results also provide a novel mechanism by which apoptotic pathways could be reactivated in advanced PCa. Employing a similar mechanism, but with opposite outcomes to PROTACs, deubiquitinase-targeting chimeras (DUBTACs) interrupt the ubiquitin mediated degradation of target proteins by removing polyubiquitin signals resulting in increased protein stability [197]. The development of a FOXO protein-targeted DUBTAC combined with PLK1 inhibition could enable greater apoptotic induction.

Seeking therapeutic interventions targeting PAX3/7-FOXO1 fusion proteins, Thalhammer et al. conducted a kinome-directed siRNA and small-molecule inhibitor screen and identified PLK1 as a regulator of PAX3-FOXO1. Xenograft models of aRMS treated with the PLK1 inhibitor, BI 2536, showed decreased expression of PAX3-FOXO1 target genes and eventually led to a complete tumor regression. Clinically, there is a strong correlation between PLK1 and PAX3-FOXO1 target gene expression, and high *PLK1* expression is associated with poor event-free survival and low overall survival [198]. These results support the notion that the PLK1/PAX3-FOXO1 signaling pathway could be a rational drug target for treating aRMS.

To date, there are no FOXK1 inhibitors reported; however, the current understanding of this protein warrants further characterization of its regulation, which may lead to novel targeted therapies.

Recent advances in drug design and delivery methods offer novel avenues for the exploration of targeted killing of cancer cells. Continued development of peptide-derived small molecules with a high affinity for target proteins, combined with our growing understanding of cryptic pockets and 3D structures, offers the potential to develop molecules with high affinity for PLK1 and FOX TFs. The optimization of these molecules around the phospho-motifs generated, or targeted, by PLK1 may enable greater specificity for neoplastic cells within which these proteins are dysregulated. Additionally, the integration of these molecules into PROTAC or DUBTAC backbones offers the potential for targeted manipulation of PLK1 and FOX TF stability. While these molecules may demonstrate potency as single agents, clinical investigation has demonstrated that combination therapies offer greater potential. As such, combinations of these proposed drugs and existing therapies must be explored for synergistic combinations.

## 5. Conclusions and Future Perspectives

Mounting evidence supports the crucial functions of PLK1 in all phases of the cell cycle and beyond. Unsurprisingly, its multi-faceted oncogenic roles across many types of human cancers continue to be uncovered. These functions are mediated by PLK1, which alters the stability, localization, and activity of hundreds of proteins, including numerous TFs, among which are FOX TFs. Both PLK1 and FOX TFs have been shown to play significant roles in human cancers, and PLK1 has been identified as an important regulator of FOX TFs, making these signaling axes promising candidates for targeted therapies (Figure 3).

Alternations in PLK1-FOX TF signaling in human cancers can be used to explore novel diagnostic and prognostic markers and inform effective therapeutic regimens. For example, the phosphorylation status of T210 in PLK1, S715 and S724 in FOXM1b, or S75 in FOXO1 could provide valuable information regarding the activity of these proteins and dysregulation of associated signaling pathways. Further characterization of PLK1-FOXO3, PLK1-FOXK1, and PLK1-FOXC2 pathways may reveal additional information for diagnostic and prognostic biomarker panel development. Recent advances in liquid biopsy approaches, particularly those employing CTCs, circulating tumor DNA (ctDNA), and, most recently, exosomes, would allow minimally invasive evaluation of the activation status of these proteins [199,200]. In addition, given the tumor heterogeneity, using quantitative single-cell proteomics, with improved workflows to reduce current costs, may also be considered [201]. Further development of these minimally invasive approaches, particularly those providing population heterogeneity information, combined with serial sampling of patients would allow effective treatment through precision medicine approaches.

Further development of PLK1 PBD-specific inhibitors and PLK1-targeted PROTACs, combined with advances in drug delivery, would allow specific targeting of PLK1 in tumors, which would restrain the proliferative, invasive, and stem cell phenotypes promoted by FOXM1 and enhance the pro-apoptotic functions of FOXOs. The combination of anti-PLK1 agents with other treatment modalities, such as FOXM1-targeting vaccines, is likely to significantly improve therapeutic efficacy. Further investigation of the regulatory role of PLK1 on FOXK1 and FOXC2 may reveal novel PLK1-FOX TF signaling cascades and previously unappreciated targets for cancer treatments. In addition, emerging technologies, such as high-throughput screenings to identify synergistic targets and next-generation sequencing integrating clinicopathologic factors to develop predictive models for patient stratification, could offer opportunities for maximizing the effectiveness of PLK1/FOX TF-targeted therapeutic strategies.

## Figures and Tables

**Figure 1 cells-12-01344-f001:**
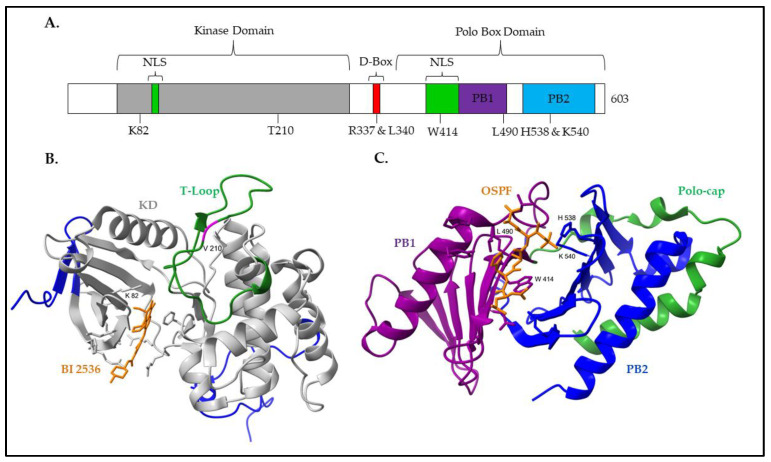
The structure of PLK1. (**A**) A schematic representation of full-length human PLK1 highlighting the major domains and amino acid residues. The kinase domain (KD) contains the key residues, K82 and T210, which are required for ATP binding and protein activation, respectively. The KD additionally contains one of the two nuclear localization sequences (NLS). On the C-terminus side of the KD is the destruction box (D-Box), which contains the key residues for proteolytic turnover of PLK1, R337 and L340. The C-terminus contains the polo-box domain (PBD), which has two polo boxes (PBs) (indicated in purple and blue, respectively). Within the two PBs, the residues that mediate interactions with substrates, W414 and L490 in PB1 and H538 and K540 in PB2, are indicated. Additionally, the schematic represents the second NLS which partially overlaps with PB1. (**B**) A crystal structure of the PLK1 kinase domain in complex with BI 2536 (orange), which is sourced from the Protein Data Bank [21,22,23]. The kinase is colored grey, the T-loop is green, and V210 is indicated in magenta within the T-loop to indicate the location of T210 in nonmutated protein. (**C**) A crystal structure of the PLK1 polo-box domain (PBD) in complex with a peptide fragment, L-H-S-pT-A, which is sourced from the Protein Data Bank [21,24,25]. PB1 and PB2 are colored purple and blue, respectively, while the polo-cap is colored green. The 3D structures were visualized and annotated using UCSF ChimeraX [26]. Note: optimal substrate peptide fragment (OSPF).

**Figure 2 cells-12-01344-f002:**
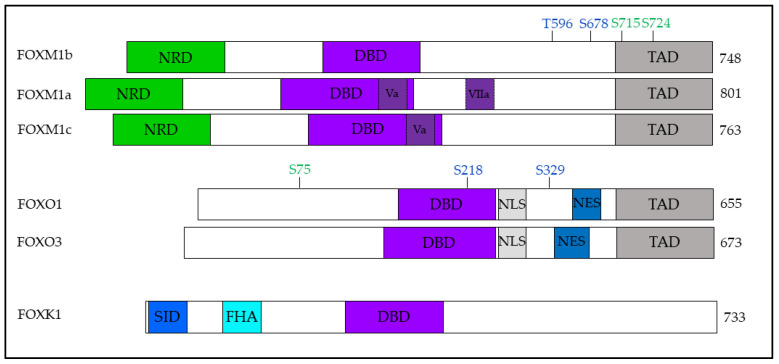
FOX TFs regulated by PLK1. Schematic representations of full-length FOX TFs with key functional and regulatory domains labeled. The most well-characterized members of the FOX subfamilies are labeled with PLK1-binding (in blue) and phosphorylation (in green) sites. Alternatively spliced FOXM1 exons, Va and VIIa, are indicated in dark purple with dashed borders. N-terminal repressive domain (NRD); DNA-binding domain (DBD); transactivation domain (TAD); forkhead-association (FHA) domain; Sin3-interacting domain (SID).

**Figure 3 cells-12-01344-f003:**
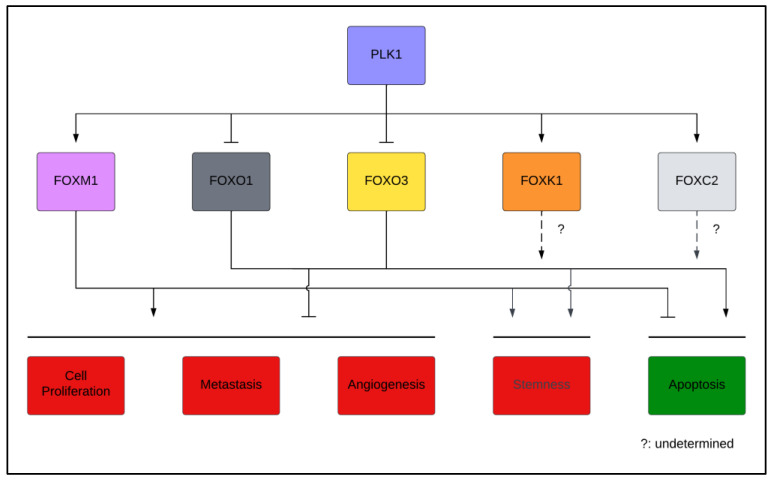
Consequences of PLK1-mediated regulation. A graphic summarizing the cancer-associated phenotypes resulting from PLK1 regulation of FOX TFs. The functional consequences of PLK1 activity upon FOXK1 and FOXC2 remain to be determined, and these uncertainties are represented by the dashed arrows with question marks. The figure prepared using Lucidchart (www.lucidchart.com accessed on 14 March 2023).

## Data Availability

Not applicable.

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
