# Peer review of "The Role of Polo-Like Kinase 1 in Regulating the Forkhead Box Family Transcription Factors"

_cells, 2023, doi:10.3390/cells12091344_

Round 1
Reviewer 1 Report
In this review, Moore et al. discussed the function role of Plk1 in regulation of Forkhead Box Family Transcription Factors and explored this regulation in cancer development and potential clinical translation. It covered most research in this field and will be very helpful for the scientists interested in Plk1 and Forkhead Box Family. Also, it will be attractive for a lot of readers.
Q1: is there a general mechanism explaining Plk1’s function in regulating Forkhead Box Family, such as FOX-DBD domain could be recognized by plk1?
Q2: are the functions or phosphorylations of Plk1 in all these Forkhead Box Family transcription factors occurred at the same spacetime or dependent on cell cycle or tissue?
Q3: the authors reviewed the important functions for these phosphorylations of Family transcription factors, how about dephosphorylation? any study focuses on the dephosphorylation of these sites?
Q4: it’s better to introduce some genetically engineered mouse work studying plk1 or Forkhead Box Family or their connections.
Author Response
Reviewer 1:
Q1: “Is there a general mechanism explaining Plk1’s function in regulating Forkhead Box Family, such as FOX-DBD domain could be recognized by plk1?”
---For the FOX TFs that have been characterized (FOXM1 and FOXO1), the regulatory phosphorylations occur in different regions of the protein. As a result, at this point in time, there does not appear to be a specific pattern for the mechanism by which PLK1 regulates these FOX TFs.
Q2: “Are the functions or phosphorylations of Plk1 in all these Forkhead Box Family transcription factors occurred at the same spacetime or dependent on cell cycle or tissue?”
---Specific information regarding the cell cycle phase and the localization of these phosphorylation has been limited to FOXM1 and FOXO1. The exact localization with which these regulatory phosphorylations occur is unclear, however, PLK1-mediated phosphorylation results in nuclear accumulation and exclusion for FOXM1 and FOXO1, respectively. Additionally, these events occur in late G2/early M-phase when PLK1 expression and activity is greatest. Currently, there is insufficient information regarding FOXO3, FOXK1, and FOXC2 to provide specific information regarding the spaciotemporal regulation of these proteins by PLK1. However, the peak of PLK1 expression and activity in late G2/early M-phase makes this period of the cell cycle a likely candidate for the point in time within which regulatory phosphorylations would occur. The revised manuscript has been edited to highlight the cell cycle phase within which the regulatory phosphorylations occur.
Q3: “The authors reviewed the important functions for these phosphorylations of Family transcription factors, how about dephosphorylation? any study focuses on the dephosphorylation of these sites?”
---The reviewer raises an interesting question. It is well known that the phosphatases PP2A and PP1 oppose PLK1 phosphorylation of mitotic substrates. However, the role of phosphatase activity in regulating the PLK1 phosphorylation sites on FOX TFs has not been investigated. This is a subject warranting future investigation.
Q4: “It’s better to introduce some genetically engineered mouse work studying plk1 or Forkhead Box Family or their connections.”
---We thank the reviewer for their suggestion. We have included genetically engineered mouse models investigating the importance of PLK1 and FOX TFs in the revised manuscript.
Reviewer 2 Report
This is a well written and informative review article, which I enjoyed reading.
In the FOX family of transcription factors, FOXM1 and FOXK1 have critical oncogenic roles in cancer, while FOXO1 and FOXO3 have broad functions in maintaining cellular homeostasis.
In this review, the authors discuss regulation of FOXs by polo-like kinase 1, which is a serine/threonine kinase. The authors also discuss the therapeutic potential of targeting PLK1-regulated FOXs in human cancers.
In many cancers, FOXM1 and p53 are reciprocally expressed. The authors should comment on this and whether p53 is also affected by PLK1.
Author Response
Reviewer 2:
“In many cancers, FOXM1 and p53 are reciprocally expressed. The authors should comment on this and whether p53 is also affected by PLK1.”
---We thank the reviewer for the insightful comments. The manuscript has been edited to incorporate a discussion of the interplay between PLK1 and p53 as well as between p53 and FOXM1.